# Diagnostic Strategies and Algorithms for Investigating Cancer Predisposition Syndromes in Children Presenting with Malignancy

**DOI:** 10.3390/cancers14153741

**Published:** 2022-07-31

**Authors:** Linda Rossini, Caterina Durante, Silvia Bresolin, Enrico Opocher, Antonio Marzollo, Alessandra Biffi

**Affiliations:** 1Pediatric Hematology, Oncology and Stem Cell Transplant Division, Padua University Hospital, Via Giustiniani 3, 35128 Padua, Italy; linda.rossini@aopd.veneto.it (L.R.); caterina.durante@aopd.veneto.it (C.D.); silvia.bresolin@unipd.it (S.B.); enrico.opocher@aopd.veneto.it (E.O.); 2Maternal and Child Health Department, Padua University, Via Giustiniani, 3, 35128 Padua, Italy

**Keywords:** cancer predisposition syndromes, tumor predisposition, hereditary malignancies, children, pediatric cancer, cancer genetics, clinical screening tool, recognition

## Abstract

**Simple Summary:**

Here we provide an overview of several genetically determined conditions that predispose to the development of solid and hematologic malignancies in children. Diagnosing these conditions, whose prevalence is estimated around 10% in children with cancer, is useful to warrant personalized oncologic treatment and follow-up, as well as psychological and genetic counseling to these children and their families. We reviewed the most recent studies focusing on the prevalence of cancer predisposition syndromes in cancer-bearing children and the most-used clinical screening tools. Our work highlighted the value of clinical screening tools in the management of young cancer patients, especially in settings where genetic testing is not promptly accessible.

**Abstract:**

In the past recent years, the expanding use of next-generation sequencing has led to the discovery of new cancer predisposition syndromes (CPSs), which are now known to be responsible for up to 10% of childhood cancers. As knowledge in the field is in constant evolution, except for a few “classic” CPSs, there is no consensus about when and how to perform germline genetic diagnostic studies in cancer-bearing children. Several clinical screening tools have been proposed to help identify the patients who carry higher risk, with heterogeneous strategies and results. After introducing the main clinical and molecular features of several CPSs predisposing to solid and hematological malignancies, we compare the available clinical evidence on CPS prevalence in pediatric cancer patients and on the most used decision-support tools in identifying the patients who could benefit from genetic counseling and/or direct genetic testing. This analysis highlighted that a personalized stepwise approach employing clinical screening tools followed by sequencing in high-risk patients might be a reasonable and cost-effective strategy in the care of children with cancer.

## 1. Introduction

### 1.1. Historical Background

The occurrence of cancer in familial clusters has been described since the sixteenth century [1]. This evidence led to the hypothesis of the existence of a genetic predisposition to develop specific tumors, which could be transmitted to descendants. The first few of these cancer predisposition syndromes (CPSs) (e.g., neurofibromatosis, multiple endocrine neoplasia syndrome, Von Hippel–Lindau syndrome etc.) were recognized and described in the literature in the beginning of the twentieth century. Affected patients often present with suggestive physical features unrelated to cancer (e.g., skin manifestation), and the diagnosis of a CPS often precedes the first occurrence of malignancy. In these disorders, the recognition of a CPS relies mostly on personal and family history and physical examination, which usually suffice for establishing the diagnosis. In the last 30–40 years, these syndromes were mapped to specific loci and the genetic cause was identified, allowing an accurate genetic diagnosis. More recently, extensive genetic studies have led to the identification of other monogenic causes of cancer predisposition without any other clinical manifestation, such as *BRCA1/2* variants in patients with breast and ovarian cancer. Affected patients often present with malignancy as first manifestation and cannot be reliably identified based on personal and family history and physical examination alone. Thus, genetic testing has gained a pivotal role in the identification of these conditions.

### 1.2. Implications of Cancer Predisposition Syndromes’ Diagnosis

Diagnosis of a CPS in cancer patients is generally pursued to warrant personalized targeted treatment, reduce toxicities, and provide clinical and psychological follow-up, as well as appropriate genetic counseling for the family [2]. The risk of developing a second malignant neoplasm is a serious issue, especially in those patients who have been treated for a malignancy in their infancy, and it is well known that CPS patients bear a much higher risk of a subsequent tumor [3,4]. However, as knowledge in the field is in constant evolution, except for a few “classical” CPS, there is no consensus about when and how to perform germline genetic diagnostic studies in children with cancer [2] While family-based germline sequencing in every child with cancer has shown potential in revealing striking information about cancer predisposition [5], this approach is not always sustainable and is often limited to research settings. The more diffused clinical practice is to refer for genetic evaluation every patient whose tumor subtype, distinctive physical features, or personal or family history raise suspicion for a CPS [2,6,7]. Various clinical screening tools have been proposed to help identify patients who carry higher risk of having a CPS [8,9,10], with heterogeneous strategies and mixed results when used in different study populations [11,12,13].

### 1.3. The Scope of Our Work

The scope of our work it to summarize the available evidence on the screening for CPSs in patients presenting with a malignancy as first manifestation through a clinical vignette employed to illustrate a prototypical CPS diagnosis. We then offer a brief overview of the historical and novel phenotypes of CPSs resulting in a higher risk of solid tumors and hematological malignancies, providing a rational approach for the screening of CPSs based on clinical manifestation. Subsequently, we review the available evidence on the overall prevalence of CPSs among unselected patients presenting with malignancy. Finally, we review the sensitivity/specificity of the clinical screening tools currently employed to identify patients at higher risk of CPS and requiring genetic sequencing, and we define the risk/benefits of employing these tools as opposed to universal sequencing.


*CLINICAL VIGNETTE: Introduction*


A four-year-old boy presents with fever, lymph nodal enlargement, and bruising. At routine blood testing, thrombocytopenia, anemia, and leukocytosis with immature cells are noted. Bone marrow shows 90% L1 immature lymphoblasts and precursor B-cell acute lymphoblastic leukemia is diagnosed based on the expression of CD19 and CD10. Common chromosomal rearrangements and aneuploidy in the leukemic clone are excluded. Thorough past personal and familial history is collected, unraveling consanguinity of the parents and a previous surgical intervention to remove a supernumerary toe.


*QUESTION: Should this patient be assessed for CPS? What are the optimal modalities?*


## 2. Prevalence of Monogenic Cancer Predisposition Syndromes

Knowledge about the prevalence of CPSs in children presenting with malignancy is critical to appropriately designing the optimal pathway to screen this population. The proportion of childhood cancer secondary to germline genetic predisposition, which was estimated to be around 4% in the ‘90s [14], grew to approximately 10% in broad sequencing studies performed in the last 10 years, probably due to the identification of previously unrecognized CPSs and the advancement of the available technologies for genetic sequencing [3,12,13,15,16,17,18,19,20]. The 10 main studies analyzing the prevalence of CPS-associated variants in children and adolescents diagnosed with cancer are shown in Table 1. These research studies employed several next generation sequencing (NGS) techniques in different cohorts of unselected patients presenting with cancer, providing data on the efficacy of universal genetic screening in this population. In most of these studies, pathogenicity of the located variants was defined with the use of population data, computational data, functional data, segregation data, ultimately classifying variants into “pathogenic,” “likely pathogenic,” “uncertain significance,” “likely benign,” and “benign”, according to the American College of Medical Genetics and Genomics and the Association for Molecular Pathology (ACMG/AMG) guidelines [21]. In those studies which did not classify variants according to the ACMG/AMG guidelines (see Table 1), criteria for variant interpretation were described in the methods. For the scope of this review, we considered of clinical interest those variants that were classified as “pathogenic” or “likely pathogenic”.

Comprehensively, this evidence shows that around 10% of children presenting with cancer can be diagnosed with a CPS by NGS.

Interestingly, this proportion did not vary substantially among different studies even if performed in diverse populations and settings. Important exceptions characterized by a lower diagnostic yield are the cohorts described by Wang et al. [15] and the cohort described by Von Stedingk et al. [19]. The study by Wang included 3006 childhood cancer survivors, showing only a 5.8% prevalence of pathogenic or likely pathogenic variants (PV or LPV) for CPSs. This is likely related to the inclusion in the study of only long-term cancer survivors and could thus further support the notion that presence of CPSs is associated to a higher mortality rate, possibly because of more severe toxicity of treatments and higher rate of secondary malignancies. Interestingly, variants in *TP53* were less frequent among cancer survivors, consistent with its lethality [15]. The Swedish cohort described by Von Stedingk et al. included 790 Swedish pediatric cancer patients, identifying PV in cancer predisposing genes in only 3.8% [19]. The lower diagnostic yield in this study could be due to the design of the gene panel that included only 22 genes, as compared to the whole exome or genome sequencing (WES or WGS) or larger panels employed in the other studies. In fact, around half of the PV identified in these studies were found in *TP53*, followed by other common genes such as *VHL*, *NF1*, *APC*, and *RB1*, while up to 30–40% of the other PV were identified in less common and less known predisposing genes. This finding highlights the genetic heterogeneity of CPSs in children and the need to include genes that have been reported in a limited number of families [18]. Moreover, most of the currently employed sequencing technologies (i.e., gene panel sequencing, whole exome sequencing) are limited in their ability to detect specific types of variants, such as variants in non-coding regions and copy number variations. This might lead to an underestimation of the prevalence of CPSs among children with cancer, and future studies employing more comprehensive technologies such as whole genome sequencing might raise our diagnostic capabilities.

## 3. Classic Cancer Predisposition Syndromes and Clinical Phenotypes Associated with a High Risk of CPSs

Classic CPSs include neurofibromatosis type 1 (NF1), Beckwith–Wiedemann syndrome (BWS), Li–Fraumeni syndrome (LFS), Von Hippel–Lindau (VHL), and multiple endocrine neoplasia (MEN) syndromes, among others. The study of the highly penetrant “classic” CPSs provides critical clues into the phenotype of patients with an enhanced susceptibility to cancer and has been of inspiration to design effective CPS screening tools. The main clinical manifestation of these CPSs include syndromic features, early occurrence of specific types of cancer and of more than one tumor in a single patient, and excessive toxic effects consequent to the use of anti-neoplastic drug regimes. Among the specific phenotypic features or congenital anomalies observed in a child with cancer that should raise suspicion of an underlying condition, facial dysmorphism and skin anomalies, neuropsychological developmental delay, growth and endocrine alterations, immune or hematological anomalies, and solid organ disfunction or malformations are particularly frequent. Classic CPSs also occur with familial clustering. We provide here a brief overview of prototypical examples of specific CPSs falling into each of these categories.

### 3.1. Syndromic Features

A typical example of a classic CPS with syndromic features is NF1, a systemic disorder with a prevalence of 1 in 3000 newborns. This condition is transmitted with an AD inheritance, with 50% of the patients carrying a de novo *NF1* pathogenic variant (PV). *NF1* acts as an onco-suppressor gene, controlling cell proliferation through the RAS signaling pathway. Penetrance is complete, with a great variability in clinical expression and symptom onset even within the same family, which makes diagnosis difficult, especially in early childhood. Disease hallmarks are multiple *café-au-lait* macules, cutaneous neurofibromas, axillary and/or inguinal freckling, and Lisch nodules (iris hamartomas). NF1 patients are prone to the development of optic pathway gliomas, benign (plexiform neurofibromas) or malignant peripheral nerve sheath tumors, central nervous system (CNS) neoplasms, and other malignancies [22]. Another CPS with overt clinical phenotype is BWS, a fairly common pediatric overgrowth disorder with an incidence of 1:10,500 births [23]. Diverse molecular defects (e.g., DNA methylation abnormalities, segmental paternal uniparental isodisomy, chromosomal abnormalities) can be identified in BWS patients, involving a cluster of imprinted genes in the 11p15.4p15.5 region; in around 20% of the affected patients, molecular diagnosis cannot be reached. The patients may present with different degrees of macroglossia, macrosomia, hemihyperplasia, and omphalocele/umbilical hernia; they carry increased risk of developing embryonal tumors, such as Wilms tumor, hepatoblastoma, neuroblastoma, rhabdomyosarcoma, and adrenocortical carcinoma [24].

### 3.2. Familial Clustering and/or Multiple Malignancies

Familial clustering of heterogenous malignancies is the hallmark of another condition predisposing to early tumors development, LFS. LFS was first described in 1969 by Dr. Frederick Li and Dr. Joseph Fraumeni Jr. after caring for four families with a peculiar pattern of cancer recurrence [25]. This AD transmitted syndrome is caused by germline PVs in *TP53*, a gene encoding for the TP53 protein that acts as a transcription factor in the control of the cellular cycle. Loss of TP53 expression predisposes to the early development of several kind of neoplasms such as brain tumors, adrenocortical carcinomas, hematological malignancies, soft tissue sarcomas and bone tumors, breast cancer, lung cancer, and others. For patients with LFS, surveillance and screening protocols have been designed to maximize early malignancies detection [26]. The same approach is essential in VHL syndrome, a condition affecting 1 in every 36,000 newborns and predisposing to the onset of different types of benign and malignant neoplasms in young adults. The patients carry a heterozygous germline PV of the *VHL* tumor suppressor gene, whose loss-of-function induces the expression of several hypoxia-inducible factors, with consequences on cell proliferation [27,28]. Malignancies typically associated to VHL disease include tumors of the CNS (retinal hemangioblastomas and cerebellar hemangioblastomas) and visceral tissues (renal cell carcinoma, pheochromocytoma, pancreatic neuroendocrine tumors). The onset of such malignancies, as well as a family history of VHL syndrome, should lead to screening for this condition to set up adequate surveillance, specific treatment, and follow up modalities [29].

Patients with synchronous or metachronous malignancies should be carefully evaluated in the suspicion of a CPS. Several factors might increase the risk of a secondary tumor in cancer patients, such as the use of specific agents (e.g., etoposide, temozolomide or radiation) and hematopoietic cell transplantation (HCT) [30,31]. Importantly, this risk is greatly increased in patients with CPS, because the CPS itself might result in metachronous precancerous lesions in several organs and several CPSs, such as Fanconi anemia (FA), Gorlin syndrome, Diskeratosis congenita (DC) and Schwachman–Diamond syndrome, might increase the treatment related genotoxicity [32,33,34]. Although in rare cases, the secondary neoplasm can affect the same organ and tissue of the first malignancy, more frequently, secondary neoplasms affect other organs. Typical secondary neoplasms include adult-type epithelial tumors (e.g., adenocarcinoma of the gastrointestinal tract, head and neck cancers, melanoma), but less frequent pathohistological subtypes are also described, especially in patients with CPSs associated to more than one type of cancer (e.g., MEN, VHL, LFS, NF1) [35]. These concepts are reflected in the finding that survivors of two cancers are more likely to carry a PV associated with CPS [15].

### 3.3. Specific or Unusual Types of Cancer

Specific cancer types are a strong predictor of an underlying CPS. For some rare malignancies, the risk of an underlying CPS can be higher than 80% in the affected patients [36]. Among the specific cancer types associated to CPS, retinoblastoma (RB), the most common ocular malignancy in childhood, which occurs in young children [37], must be kept in mind. Approximately 40% of all RB are due to germline PV in *RB1* tumor-suppressor gene, with about 80% of them arising *de novo* in the absence of a clear family history [38]. Children with hereditary RB may develop bilateral RB and less frequently midline intracranial primitive neuroectodermal tumors (PNET) (defined as trilateral RB when occurring in addition to mono or bilateral RB); moreover, they carry increased risk of developing a secondary primary malignancy as compared to patients without germline PV in *RB1* [39].

Colon cancer, despite being less frequent in children, may also be the first manifestation of a CPS. Traditionally, CPS associated with gastrointestinal cancer are divided in those associated with gastrointestinal polyposis (i.e., familial adenomatous polyposis, MUTYH-associated polyposis, Peutz–Jeghers syndrome), predisposing to colorectal cancer and other malignancies since childhood (hepatoblastoma, medulloblastoma, ovarian sex cord stromal tumors, and Sertoli cell testicular tumors, among others) and those with a smaller number of polyps and a predominantly cancer phenotype [40]. The latter essentially includes Lynch syndrome, determined by monoallelic PV variants in the mismatch repair genes, which presents with a distinct phenotype from its biallelic form, the constitutional mismatch repair deficiency (CMMRD) syndrome [41]. LS is characterized by gastrointestinal and genitourinary malignancies that usually present in adult age, but with increasing reports of pediatric onset neoplasms [42].

CNS tumors can also be associated with predisposing conditions, including Choroid plexus carcinoma (CPC), high grade glioma (HGG) or SHH-activated medulloblastoma in LFS and HGG in the CMMRD syndrome [43]. Other examples of frequently associated malignancies and syndromes are juvenile myelomonocytic leukemia (JMML) in patients with NF1 and Noonan syndrome [44,45]; hypodiploid acute lymphoblastic leukemia (ALL) and choroid plexus carcinoma in children with LFS [46,47]; medullary thyroid carcinoma (MTC) and multiple endocrine neoplasia type 2 (MEN2) [48]; pleuropulmonary blastoma and *DICER1* PV [49,50]; pheochromocytoma and VHL, NF1, MEN2 [29,51]; pheochromocytoma and paraganglioma in *SHDx*-mutation carriers [52].

Among classic CPSs, MEN syndromes, a heterogeneous group of conditions with a distinct spectrum of benign and malignant manifestations involving the endocrine glands, are to be mentioned. Since MEN1 was first depicted in 1903, to date, three main syndromes have been identified (MEN1, MEN2, MEN4), all transmitted with an AD inheritance [48]. The most frequent is MEN1, with an incidence of 1:30,000 births. Loss of function variants in the *MEN1* tumor suppressor gene cause altered functioning in endocrine tissues (most frequently hyperparathyroidism) and tumorigenesis, especially involving parathyroid glands, pituitary glands, endocrine pancreas, and less frequently non-endocrine organs and tissues [53]. MEN2 is caused by pathogenic variants of the *RET* gene. Patients may present with variable associations of MTC, pheochromocytoma, primary hyperparathyroidism, cutaneous lichen amyloidosis, and Hirschprung’s disease (MEN2A), or MTC, phaeochromocytoma, ganglioneuromatosis of the gastrointestinal tract, and musculoskeletal and ophthalmologic abnormalities (MEN2B) [54]. Those patients in families presenting with medullary thyroid carcinoma-only are classified as having familial MTC, a distinct subtype of MEN2A [55].

The last MEN syndrome discovered is MEN4, where PVs in the tumor suppressor gene *CDKN1B* increase the susceptibility of developing primary hyperparathyroidism and pituitary adenomas [56].

### 3.4. Excessively Toxic Effect of Treatments

Excessive toxicity consequent to anticancer regimes is another red flag for CPS. Predisposition to cancer and sensitivity to ionizing radiation are distinctive features of DNA repair syndromes, a genetically heterogeneous group of disorders including ataxia telangiectasia, Bloom syndrome, Nijmegen breakage syndrome, FA, Schimke syndrome, and DC [57,58]. Patients with FA and DC are, in addition, at risk of increased toxicity when exposed to alkylating agents. Affected patients suffer very severe mucosal and organ damage and prolonged cytopenia after being exposed to standard doses of certain cytotoxic drugs. Genetic testing is thus crucial to diagnose these conditions in patients with exaggerated treatment-associated toxicity, while considering that several functional tests are not feasible in patients that receive cytotoxic agents or blood transfusions (e.g., the diepoxybutane test for FA). A rapid diagnosis is relevant for the management of the patients, as specific reduced-intensity treatment protocols are available for several DNA repair syndromes and HCT might be an unavoidable step for some cancers. It is also well known how patients with trisomy 21 suffer increased methotrexate (MTX) toxicity, and treatment protocols usually include adjusted MTX doses for Down patients [59]. Interestingly, in CMMRD, another disorder affecting DNA repair, normal tissue response to genotoxic agents and radiotherapy is maintained but, as adequate mismatch repair is required for the efficacy of some chemotherapeutic agents (e.g., mercaptopurine, temozolomide), the tumor is less responsive to these agents [41]. Thus, patients with CMMRD also require personalized treatment plans taking account the peculiarity of the condition.

## 4. Cancer Predisposition Syndromes Specifically Associated with Hematologic Malignancies

### 4.1. Specific Characteristics of Cancer Development in the Hematopoietic System

Tumorigenesis in the hematopoietic systems has several specificities due to its development and physiology throughout life. Firstly, the functioning of the adaptive immune responses is based on an extensive somatic recombination of nuclear DNA, which is unique to the hematopoietic system. For example, RAG1/2 DNA recombinases are critical for the development of the immunoglobulin and T-cell receptor repertoire, but might also affect other DNA regions, causing breakpoints that result in chromosomal translocations driving leukemogenesis [60]. Secondly, the very short life of several blood cell types (e.g., neutrophils and platelets) requires a very active and tightly regulated replication of the hematopoietic stem and progenitor cells that persists throughout the whole life. The number of stem cell divisions is a crucial driver of tissue specific tumors [61]. Finally, the study of hematologic malignancies has pioneered the description of genetic mechanisms resulting in neoplastic transformation that could be leveraged for therapeutic intervention. The first clearly defined genetic lesion associated with leukemia that resulted in the development of an efficacious treatment was the *BCR-ABL* translocation in chronic myeloid leukemia, leading to the development of the tyrosine kinase inhibitor imatinib in the 1990s [62]. Easy accessibility of malignant cells facilitated research and therapy design and is currently reflected by the very wide use of various forms of genetic characterization of the tumor cells in these disorders. Frequently, NGS sequencing of the cancer cells aimed at identifying potentially targetable somatic lesions may also shed light on an underlying germline disorder [63]. Indeed, as with several other tumors, genes harboring germline variants associated with cancer predisposition are the genes that are most frequently somatically mutated in the same type of neoplasia [64].

### 4.2. Predisposition to Myeloid Malignancies

Although several genetic disorders such as LFS and FA can predispose to both solid tumors and hematological malignancies, there is a wide range of disorders mostly related to a high risk of leukemia and lymphoma. More specifically, genetic defects are often mainly associated with a single subtype of hematopoietic cancer, such as acute myeloid leukemia (AML), or acute lymphoblastic leukemia (ALL), or B-cell lymphoma, with minor overlap between these disorders. An example of such specificity is provided by patients harboring biallelic *loss-of-function* variants in *ERCC6L2* who are exquisitely predisposed to acute erythroid leukemia (AML M6) with acquired somatic variants in *TP53* [65]. Predisposition to myeloid and lymphoid malignancies is mostly associated with variants in genes encoding for key transcription or regulatory factors implicated in the lineage differentiation. AML or MDS are often associated with *loss-of-function* variants of *GATA2*, a gene involved in the development and maintenance of the hematopoietic stem cells [66,67]. Moreover, *SAMD9/SAMD9L* variants predispose to refractory cytopenia and MDS, with a propensity for somatic rescue. These genes are linked to tumor suppression, inflammation, development, and protein translation. Both mutated genes were reported with a prevalence of about 7–8% and are strictly associated with monosomy of chromosome 7 [68]. In MDS, *GATA2* is predominant in high risk (i.e., RAEB/RAEB-t) patients, while *SAMD9/SAMD9L* is inherent on RCC and hypocellular BM [68]. Moreover, other disorders resulting in an increased stress on granulopoiesis, such as Schwachman–Diamond syndrome (due to variants in *SBDS* and others) or severe congenital neutropenia (Kostmann syndrome, due to variants in *ELANE*, *HAX1*, and others) are associated with an increased risk of myelodysplasia and myeloid leukemia [69]. A recent study on a large number of unselected adult patients with AML found that almost 14% of the patients carried pathogenic or likely pathogenic germline variants in genes predisposing to myeloid malignancies, without a specific association with family history or other clinical factor such as type of leukemia manifestation or gender [70]. Germline variants can confer a high risk of developing not only AML, but also myeloproliferative disorders. The classical example of such occurrence is Noonan-like syndrome due to heterozygous variants in *CBL*. These patients have a high risk of developing a rare myelodysplastic (MDS)/myeloproliferative neoplasm (MPN) overlap syndrome, known as juvenile myelomonocytic leukemia (JMML). The development of this clonal disorder is associated with the loss of heterozygosity (LOH) at the *CBL* locus. The clinical course of *CBL*-associated JMML might be benign, without the need for treatment in some patients. Thus, the genetic dissection of CPS in this setting might have immediate consequences on management.

### 4.3. Predisposition to Lymphoid Malignancies

Conversely, predisposition to lymphoid tumors is frequently interlaced with other disorders of the adaptive immune system resulting in inborn errors of immunity (IEI). This has been described both for ALL and for mature B-cell lymphoma. Germline mutations in genes such as *PAX5*, *ETV6*, and *IKZF1* predispose to ALL and their recognition is mandatory for clinical management, given the high risk of developing these tumors. Indeed, somatic *IKZF1* deletions and mutations occur frequently as somatic alterations in B-ALL [71] and Churchman et al. reported a heterozygous germline variant with an autosomal dominant transmission resulting in predisposition to childhood ALL [72]. Furthermore, pivotal hematopoietic genes are reportedly mutated at the constitutional level in patients with T-ALL; recently, we reported a novel variant in *PIK3R1* in a patient with Short syndrome and *TAL/LMO2* T-ALL, suggesting that this might be a novel locus resulting in predisposition to T-ALL [64].

### 4.4. Inborn Errors of Immunity and Risk of Hematologic Malignancy

Several mechanisms provide insight into the existing link between a higher risk of hematologic malignancy and IEI affecting the adaptive immune system, which has been described both for monogenic disorders (e.g., *SH2D1A* deficiency) and phenotypically-defined disorders (e.g., common variable immune deficiency) [73,74]. Defects in the adaptive immune system pose a substantial stress on the development of T and B cells that often can lead to an un-discriminated lymphoproliferation [75]. Several patients present with an abnormal accumulation of lymphocytes in secondary lymphoid organs or other tissues, resulting in splenomegaly, diffuse lymphadenopathy, and organ damage [76]. Moreover, the defect in the immune system results in a weakened immune surveillance of pre-cancerous cells that are less efficiently controlled by cytotoxic T cells. Finally, IEI can affect pathways implicated in the regulation of cell division (e.g., PIK3 kinase pathway) or DNA stability (e.g., ataxia telangiectasia) and disruption of these factors is a major oncogenic driver [77].

A peculiar example of an IEI affecting both the innate and adaptative immune response and resulting in a higher risk of cancer is *GATA2* haploinsufficiency. This transcription factor is crucial to maintaining the appropriate number and function of CD4+ T cells, B cells, monocytes, and NK cells, beyond other non-hematopoietic tissues. The loss of such a broad function results in very diverse and heterogenous phenotypes, including a broad range of infections, cytopenia, and pulmonary, vascular, and audiological manifestations [78]. However, the risk of hematological malignancies in patients with GATA2 deficiency is mostly limited to AML and MDS, due to its cell-specific biological effect [79,80].

## 5. Clinical Tool Validation Studies

The relationship between cancer and phenotypical abnormalities has been subject of study for decades to delineate the role of genetic factors in tumorigenesis and to prioritize patients for genetic testing. Children with cancer have a significantly higher prevalence of morphological anomalies that are even more frequent among patients with CPS [81]. However, not all CPS patients have an overt clinical phenotype and more accurate evaluation is needed to assess the risk of CPS [16]. In the last decade, several clinical screening tools have been proposed to identify children who are at risk of having a CPS and need to undergo genetic evaluation. Such tools leverage the specific characteristics of the CPSs we described in the previous chapter and are mainly based on the presence of morphological abnormalities, personal and family history, type and number of tumors, and age of occurrence. The three most studied clinical screening tools are: the Childhood Cancer Screening checklist (CCSC) [8,11]; the Jongmans’ criteria (JC) in their original [10] and updated version (Jongmans’ modified criteria or JMC) [51]; and the McGill Interactive Pediatric Oncogenetic Guidelines (MIPOGG) [82]. They are represented in Table 2.

### 5.1. The Childhood Cancer Screening Checklist

In the algorithm proposed by Hopman et al., a list of CPS crucial manifestations was associated to the use of 2D and 3D photographs [83]. With the limitation of the small size of the cohort and the presence of a selection bias, this tool based on physical examination allows clinicians to identify more patients needing genetic referral then the regular consultation. This work pioneered the development of the CCSC, implemented with the addition of questions regarding tumor type and personal and family history. With these modifications, the instrument showed a sensitivity of 100%, but measured a much lower prevalence of CPS than expected (1%), mainly because of the exclusion of patients already having a known CPS diagnosis [11].

### 5.2. Jongmans’ Original and Modified Criteria

While the CCSC specifically focuses on the morphologic evaluation of the patients (as described in Table 2), family history and the type of malignancy are the fundamental criteria of the most widely used screening tool at present, the Jongmans’ criteria. This clinical selection tool also takes into consideration the presence of multiple primary malignancies, specific congenital anomalies, and excessive toxicity as a consequence of cancer therapy [10]. Notably, JC is the only tool to advise for genetic counseling those patients who had suffered of excessive treatment toxicity and whose tumors have genetic defects suggestive of CPS germline predisposition (i.e., hypodiploid ALL in LFS).

Ripperger et al. validated and implemented JC by administering the modified version of the Jongmans questionnaire developed by the cancer predisposition working group of the Germany Society of Paediatric Oncology and Haematology (Jongmans’ modified criteria, JMC, Table 2) [51]. The use of this updated version, including a wider list of CPS associated tumors, allowed the identification of significantly more patients with CPS compared to the control group, with a proportion of children diagnosed with a CPS similar to that identified by genetic testing [16]. The authors claim that this approach allows the preferential identification of children with a clinically relevant CPS, while sequencing all pediatric cancer patients is thought to increase the probability of identifying variants of unknown clinical significance.

In a Turkish study by Demirsoy et al., JC was applied with the additional inquiry of detailed cancer family history up to third degree relatives. In this study, a relatively high ratio of candidate CPS patients were identified, but only a minor proportion of them were confirmed to have a CPS [84]. This result may show lower selection power of JC in populations with high rate of consanguineous marriages, since consanguinity was the most frequent indication for genetic referral in the Turkish cohort.

A German historical case-control study assessing the prevalence of CPS diagnosis with or without the use of JMC on newly diagnosed pediatric cancer patients confirmed the potential of this tool for the identification of clinically relevant CPSs; during the study period, 9.4% were diagnosed with a CPS versus 5.3% in the historical control group (*p* = 0.032) [85].

### 5.3. The McGill Interactive Paediatric OncoGenetic Guidelines

Another well studied clinical selection tool is the MIPOGG, a Canadian evidence-based support tool consisting in algorithms including “universal criteria” (high-risk features strongly suggesting the presence of a CPS) and “tumor-specific criteria” [82]. This tool is interactive and freely available at https://mipogg.com, (accessed on: 1 June 2022). Its validation in three retrospective populations of children with diverse oncologic conditions (ovarian tumors, Wilms tumor, and other malignancies) highlighted a sensitivity of 100% [9,86] and the potential to decrease time to CPS recognition [87]. Interestingly, the MIPOGG also had some utility in identifying cancer survivors at increased risk of a second malignancy in a Canadian case control-study, especially among patients with non-hematological malignancies who were not exposed to radiation [88].

### 5.4. Comparing Clinical Screening Tools and Genetic Testing

Overall, the results of most of these studies conducted both in research and clinical settings support the role of the screening tools in addressing more patients to the genetic testing than the standard clinical evaluation [9,10,11,83,84,85,86,87] (Table 3).

To provide a clear measure of sensitivity and specificity, several studies investigated the use of clinical selection tools paired with genetic analysis. Byrjalsen et al. investigated a cohort of 198 Danish children with cancer with a 14.6% prevalence of genetically verified CPSs [12]. In this study, 70 out 198 patients (35.4%) were suspected of having a CPS according to the Jongmans’ or MIPOGG screening tools, and only 4 patients with confirmed PV were missed by usage of both tools. Despite the suboptimal specificity, screening tools have a good sensitivity. In a recent paper by Gargallo et al., 9.4% of the patients of a pediatric oncology unit were diagnosed with a genetically verified CPS [19]. JC was evaluated during the workflow, using it to address specific testing if a depicted CPS was suspected but nonetheless testing the entire cohort. This allowed the validation of a sensitivity and specificity respectively of 94% and 77% when using the classic JC, with the sensitivity reaching 100% when using the JMC by Ripperger [51].


*CLINICAL VIGNETTE: Conclusion and relevance*


Jongmans’ modified criteria were applied to assess the need for CPS screening: consanguinity and skeletal abnormalities are independent indications to genetic evaluation. Thus, NGS CPS panel sequencing was performed, revealing a large homozygous deletion in both alleles of *FANCA*, which was confirmed by aCGH. The parents were heterozygous carriers of the deletion. A diagnosis of Fanconi anemia was thus made, implying the need for a personalized treatment for the patient to avoid excessive toxicity. Unfortunately, despite initial response to intensive chemotherapy with avoidance of high dose alkylating drugs, the patient suffered several complications (severe mucositis, bacterial sepsis, pulmonary aspergillosis, prolonged aplasia, steroid-induced diabetes mellitus) and ultimately died of relapsed leukemia. Her siblings’ genetic evaluation revealed the same genetic aberration in a younger brother, who is currently in undergoing the surveillance according to the most current recommendations. The parents, who are willing to have more children, have been informed about the recurrence risk and appropriate genetic prenatal and preimplantation management is offered.

## 6. Conclusions

Whilst determining which children could be carrier of a CPS-associated variant remains a major challenge for the pediatric oncologists, the rapidly and constantly changing landscape in the field of hereditary cancer suggests that a careful evaluation of the optimal diagnostic strategy is paramount to providing timely recognition of CPS and at the same time avoiding unnecessary testing and uncertain results. The currently available approaches include universal genetic screening or testing of patients only when fulfilling appropriate clinical criteria. We synthetized the pros and cons of these approaches in Figure 1. 

Extended genomic sequencing programs, including pan-cancer studies and trio WES sequencing, provide insightful information on cancer pathogenesis and targeted treatment [5,17,18,89,90], but the full analysis of their results requires significant resources and long-term follow-up, promising benefits that may not be realized for many years yet [3], and their use is usually limited to research setting. The benefit of a universal genomic approach may be the identification of recently described and little known CPS-associated variants [18], potentially unraveling new associations between tumor subtypes and CPS genes and uncovering new CPS genes [13]. However, we must keep in mind that identification of variants in CPS genes does not inevitably imply causality; moreover, broader sequencing is associated with the discovery of increasing numbers of variants of unknown significance, including variants associated with the development of cancer in adult age or with health implications other than cancer [3,16,19]. On the other side, clinical screening tools provide valuable information, increasing the pre-test probability of diagnosing CPS. They are easy-to-use tools with excellent sensitivity and a sufficient specificity (especially when combined [13,91]) and can be employed in daily clinical practice to reduce costs and avoid unnecessary genetic testing.

In conclusion, we believe that a stepwise approach with the initial application of well validated screening tools, followed by targeted analysis with or without deeper sequencing, would be a reasonable and cost-effective strategy in the care of children with cancer. This approach is sufficiently sensitive, contains costs, and restricts undesired or difficult-to-interpret findings, protecting the already struggling families from additional anxiety and disease burden. Appropriate pre- and post-test counseling needs to be carried out when applying the screening tool and then again when proposing genetic testing. When sequencing, we believe that the use of screening panels including a large number of genes should be preferred, given the diversity of genetic causes of CPS and unexpected genotype–phenotype associations. The clinical screening and genetic testing would be carried out early in the care of cancer patients, as the results might have immediate implications for the surveillance and, in some cases, the treatment of the patients, and testing of long-term survivors might introduce a substantial bias due to early mortality of patients with CPS. Further studies are needed to determine the feasibility and accuracy of this approach in wider populations of children with cancer.

## Figures and Tables

**Figure 1 cancers-14-03741-f001:**
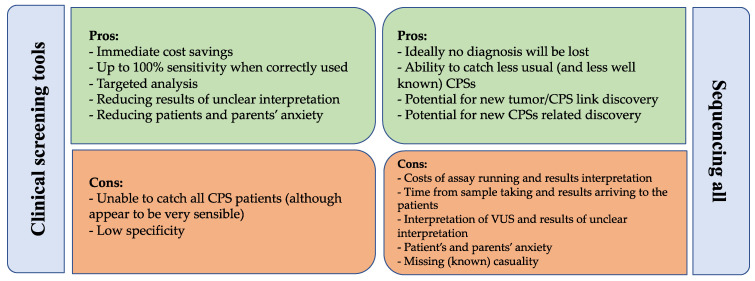
Discussion of pros and cons of screening for cancer predisposition syndrome in children with cancer with a clinical screening tool or with universal sequencing. Abbreviations: CPR—cancer predisposition syndrome. VUS—variant of unknown significance.

**Table 1 cancers-14-03741-t001:** CPS prevalence studies.

Reference	Zhang, N Engl J Med 2015	Grobner, Nature 2018	Wang, Journal of Clinical Oncology 2018	Parsons, JAMA Oncology 2018	Chan, Npj Genomic Medicine 2018	Byrjalsen, PLoS Genetics 2020	Gargallo, Cancers 2021	Von Stedingk, Scientific Reports 2021	Fiala, Nature Cancer 2021	Newman, Cancer discovery 2021
**Setting**	St. Jude Children’s Research Hospital, USA	Multicenter international, German based	St Jude Children’s Research Hospital, USA	Texas Children’s Hospital, Houston, USA	National Cancer Centre, Singapore	Multicenter national, Denmark	Valencia University Hospital, Spain	Multicenter, southern Sweden	Memorial Sloan Kettering Cancer Center, New York, USA	St. Jude Children’s Research Hospital, Memphis, US
**N°**	1120	914 pts (961 tumors)	3006	150	102	198	170	790	751	300
**Patients selection**	Cancer patients <20 yo	Cancer patients 95% <18 yo and 5% young adults up to 25 yo	Childhood cancer survivors (>5 yo since diagnosis, >18 yo)	Children with newly diagnosed CNS or non-CNS solid tumors	Cancer patients <18 yo	Newly diagnosed cancer patients aged 0–17 yo	Cancer patients 0–18 yo	Cancer patients <18 yo at diagnosis	Patients with solid tumors	Children with newly diagnosed (85%) ore relapsed/refractory (15%) cancers
**Type of sequencing**	595 WGS, 456 WES, 69 both	547 WGS, 414 WES on tumors and matched germline samples	WGS	WES on tumor and germline samples	WES, MLPA	WGS	Specific gene testing and/or Custom NGS panel (OncoNano V2)	Targeted sequencing of 22 CPS genes	NGS panel of 468 genes	WES/WGS on tumor and germline samples
**Comparison with clinical tools**	no	no	no	no	JMC, CSCS	JC, MIPOGG	JC	no	no	no
**Results: clinical tools**	/	/	/	/	Specificity: 24% JMC, 38% TuPS, 52% when combined Sensitivity 100% alone or combined	47.5% carried PV in a CPS gene or were suspected of having a CPS based on JC/MIPOGG	94% sensitivity, 77% specificity	/	/	/
**Results: genetics**	PV or LPV in 8.5%	Germline PV in 7.6%	PV or LPV in 5.8%	10% carried PV or likely PV related to their phenotype; 6% (n = 10) were found to have single PV associated with AR CPS (only 1 had a tumor type associated with the AR condition)	PV in 9.8%	14.6% carried PV in at least 1 CPS gene (10.6% childhood onset CPS, 4.5% adult onset CPS)	PV in 9.4%; 5.9% likely PV	PV in 3.8%	PV in 13% (moderate/high penetrance AD genes); PV in 18% (low/moderate/high penetrance AD or AR genes)	PV or LPV in 18% of 300 patients
**Notes**	Only 40% of mutated patients have family history of cancer	Correcting for the relative incidence of cancer types, the predicted frequency germline PV in cancer patients is 6%	Only alive patients were sequenced: variants associated with increased mortality risk were underrepresented; survivors with SMN were more likely of having a CPS	15 patients brought a PV or a LPV underlying the phenotypic presentation, but only 5/15 were found to be genetically testing, as decided by the team caring for the patient	In patients harbouring more than one pathogenic germline mutation, clinical manifestations were predominantly consistent with genes in which penetrance is greater at an earlier age	4 mutated patients did not fulfil clinical screening tools criteria	Only 1 patient was "lost" by JC; JMC in this cohort would have had 100% sensibility	On an individual gene basis, the difference between this and the Zhang and Gröbner cohorts was the prevalence of TP53 mutations	Individuals who tested positive for a P/LP variant were more likely than those who tested negative to have had multiple primary cancer diagnoses (10% versus 3%)	55% of germline PV/LPV was considered relevant to tumor formation (that is, there was a known association between gene and tumor type or a specific molecular evidence supporting a functional consequences of the mutation in the tumor)
**Variants classification according to ACMG/AMP criteria**	Yes	No	Yes	No	Yes	Yes	Yes	Yes	Yes	Yes

Abbreviations: CPS cancer predisposition syndrome. WGS—whole genome sequencing. WES—whole exome sequencing. NGS—next generation sequencing. MLPA—multiplex ligation-dependent probe amplification. PV—pathogenic variant. LPV—likely pathogenic variant. —secondary malignant neoplasm. CNS—central nervous system. AD—autosomal dominant. —autosomal recessive. JC—Jongmans’ criteria. JMC—Jongmans’ modified criteria. CCSC—childhood cancer screening checklist. CCSC—childhood cancer screening checklist. ACMG/AMP—American College of Medical Genetics and Genomics and the Association for Molecular Pathology guidelines.

**Table 2 cancers-14-03741-t002:** Clinical screening tools.

	Clinical Tools			Evaluated by:
	CCSC	JMC	MIPOGG	
**Family history**	≥2 times the presence of the same specific kind of cancer (on one side of the family till the third degree), which could be associated with the malignancy of the child	≥2 malignancies occurred in family members before age 18 years, including index patient	Known cancer predisposition syndrome in the family	All three tools
	Another family member with childhood cancer (≤18 y.o.), which could be associated with the malignancy of the child	Parent or sibling with current or history of cancer before age 45 years	Close relative * with cancer <18 years OR a parent/sibling/half-sibling with cancer at <50 years	
	≥2 family members (on one side of the family till the third degree) with cancer <45 y.o., which could be associated with the malignancy of the child	≥2 first or second degree relatives in the same parental lineage with cancer before age 45 years	Close relative * with the same cancer type or same organ affected by cancer at any age	
	A first degree family member of this child (parent, sibling) with cancer has (had) cancer themselves	The parents of the child with cancer are consanguineous	Close relative * with multiple primary tumors	
**Tumor type**	Rare tumor, specific malignancy at unsuspected age, unusual location. See also the list §	Neoplasm indicating CPS §	Tumors for direct referral §	All three tools
**Genetic tumor analysis**		Genetic tumor analysis reveals defect suggesting a germline predisposition		JMC only
**Previous history**	Prior primary malignancy	A patient with ≥2 malignancies	>1 primary tumor	All three tools
	Perinatal data, learning and developmental difficulties, or growth failure possibly existing in the context of a CPS		Bilateral/multifocal primary tumor	
	Other medical issues possibly existing in the context of a CPS			
**Phenotypical examination**	Abnormalities in the appearance suggestive for a CPS	Congenital anomalies	Dysmorphic features/congenital abnormalities that the clinician deems to be related to cancer predisposition	Morphologic abnormalities better defined in CCSC
	a. Found during physical exam (checklist)	Facial dysmorphism		
	* Head: scalp tumors, brittle hair	Mental impairment, developmental delay		
	* Eyes: cataract, visible nerve fibers on cornea, photosensitivity	Abnormal growth		
	* Ears: crease/pits of ear lobule, helical pits of ear helix	Skin anomalies (Abnormal pigmentation such as ≥2 *café-au-lait* spots, vascular lesions, hypersensitivity to sun, benign tumors)		
	* Mouth: leukoplakia, abnormal tongue, oral pigmentation, oral tumors, abnormal oral mucosa, mucosal neurinomas, papilloma peri-orificial	Hematological abnormalities (not explained by current cancer)		
	* Thorax: supernumerary nipples	Immune deficiency		
	* Abdomen: umbilical hernia	Endocrine anomalies		
	* Extremities: asymmetry, palmar pits			
	* Genitalia: abnormal pigmentation, ambiguous genitalia			
	* Skin: teleangectasia, skin tumors, blue nevus, axillary freckling, hyperpigmentation, thin skin/generalized skin atrophy			
	* Neurological: ataxia, cranial nerve palsy			
	* Endocrine: enlarged thyroid			
	b. 2D photographic series			
	c. 3D photograph			
**Treatment toxicity**		The patient suffers from excessive toxicity of cancer therapy		JMC only
**Tumor specific algorithm**		Yes	Yes	MIPOGG only

Abbreviations: CPS—cancer predisposition syndrome. CCSC—childhood cancer screening checklist. MIPOGG—McGill Interactive Pediatric Oncogenetic Guidelines. JMC—Jongmans’ modified criteria. * Close relative: parent, sibling, aunt/uncle, first cousin, grandparent. § The list of malignancies requiring direct genetic referral according to CCSC, JMC, and MIPOGG is reported in Appendix A.

**Table 3 cancers-14-03741-t003:** Results of studies evaluating the use of clinical screening tools.

Study	Hopman, European Journal of Cancer 2013	Postema, Familial Cancer 2021	Goudie, Pediatric Blood and Cancer 2018	Goudie, Pediatric Blood and Cancer, SIOP19 Abstract	Cullinan, International Journal of Cancer 2020	Cullinan, Journal of Clinical Oncology 2021	Schwermer, Familial Cancer 2021	Demirsoy, European Journal of Medical Genetics 2021
**Setting**	Multicenter international, Netherlands based (pilot)	Multicenter national, Netherlans	Multicenter, Canada	Multicenter, Canada	Multicenter, Canada	Multicenter, Canada	Hannover Medical School, Germany	Kocaeli University Department of Pediatric Oncology, Turkey
**Number of participants**	10	363	278	422	180	1886	739	123
**Patients**	Children with a newly-diagnosed cancer already evaluated by a geneticist	Children with a newly diagnosed neoplasm (malignant, benign or borderline) without a known CPS diagnosis	Children with neuroblastic tumor	Children 0–18 yo with a newly-diagnosed cancer and a confirmed CPS	Patients <18 yo treated for Wilms tumor	Childhood cancer survivors diagnosed or treated before 18 yo who developed a SMN (cases) or did not (controls)	Children with a newly-diagnosed cancer	Children 0–18 yo with solid tumors
**Screening tool**	49 scored manifestations of CPS	CCSC	MIPOGG	MIPOGG	MIPOGG	MIPOGG	JMC	JC + cancer family history up to 3rd generation
**Characteristics object of the tool**	Perinatal history, family history and physical examination (including 2D and 3D pictures)	Perinatal history, family history and physical examination (including 2D and 3D pictures)	Universal criteria (personal and familial history) and tumor-specific criteria	Universal criteria (personal and familial history) and tumor-specific criteria	Universal criteria (personal and familial history) and tumor-specific criteria	Universal criteria (personal and familial history) and tumor-specific criteria	Family history, CPSs related neoplasms, >1 malignancy, morphologic anomalies, excessive toxicity	Family history, CPSs related neoplasms, >1 malignancy, morphologic anomalies, excessive toxicity
**Workflow**	Geneticists in regular consultations vs geneticist with the screening instrument	8 CGs indicating referral or not based on the tool; 1/3 of the pts for whom referral was not indicated were the control group	2 coinvestigators applied the tool to the clinical data	2 indipendent clinicians applied the tool	Retrospective application of the tool	Retrospective review of patients data for MIPOGG application and subsequent case control comparison	287 pts (2017–2019) were administered the JMC and tested if indicated by the subsequent genetic evaluation; 452 pts (2012–2016) served as controls	Interview and data collection
**Results**	Geneticists using the instrument deduced more reasons for referral than the geneticist that judged based on the regular consultation	Sensitivity 100%, specificity 43% but CPS prevalence 1%	Agreement 83% between algorithm and physicians (+15 patients identified by the algorithm alone)	Sensitivity 99.3%	Sensitivity 100%	A MIPOGG output recommending evaluation was significantly associated with SMN development (HR 1.53; 95% CI, 1.06 to 2.19)	9.4% were diagnosed with a CPS in the JMC group against 5.3% in the controls (P = 0.032)	28.8% had indication for genetic referral according to JM, rising to 42.3% when considering 3rd generation family history
**Notes**			All 6 children with confirmed CPS were identified by MIPOGG as needing genetic referral					

Abbreviations: CPS—cancer predisposition syndrome. CCSC—childhood cancer screening checklist. CG—clinical geneticist. MIPOGG—McGill Interactive Pediatric Oncogenetic Guidelines. JC—Jongmans’ criteria. JMC—Jongmans’ modified criteria. SMN—secondary malignant neoplasm.

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
