# Peer review of "Diagnostic Strategies and Algorithms for Investigating Cancer Predisposition Syndromes in Children Presenting with Malignancy"

_cancers, 2022, doi:10.3390/cancers14153741_

Round 1

Reviewer 1 Report

Dear Editor-in-chief,

The authors adequately addressed all the raised issues and the paper is now worth publishing.

The paper is a significant contribution to the field.

Author Response

Dear Reviewer,

Thank you for your comments. We are glad that you appreciated our work.

Sincerely, 

Linda Rossini

Reviewer 2 Report

Thank you very much for revising the manuscript. I have no further comment. However, the uploaded version is hard to read as several edited paragraphs contain the initial sentences as well as the revised parts; this needs to be corrected.

Author Response

Dear Reviewer,

Thank you for your comments.  In the revised version, we deleted the initial sentences.

Sincerely,

Linda Rossini

This manuscript is a resubmission of an earlier submission. The following is a list of the peer review reports and author responses from that submission.

Round 1

Reviewer 1 Report

Rossini et al. provide a review regarding the diagnostic strategies and algorithms for investigating CPS in children diagnosed with a malignancy. The authors provide a nice overview discussing several critical and interesting issues in that regard. Please find below my major and minor concerns and comments.

Major concerns

General

The authors should clearly distinguish between in their argumentation between diagnostic approaches and research settings.

It needs to clearly stated throughout the manuscript that the identification of genetic variants in CPS genes (i) requires proper classification regarding their clinical impact using current general guidelines and gene specific recommendations such as ACMG/AMP criteria and, if available, their ClinGen specification and (ii) does not immediately imply causality. These basic principles have unfortunately not been applied in all cited/discussed (sequencing) studies and we need to keep this in mind. The authors should state somewhere that their abbreviation PV includes (likely) pathogenic variants (i.e., class IV and V).

The authors mainly focus on the screening tools. However, they should discuss the clinically critical question of “What to test?” in more detail. In the final paragraph, the authors recommend the use of CPS gene panels but this recommendation should be addressed in more detail and again need to keep in mind the differences between diagnostics and research.

The authors may consider to concise their review by focusing on their main topic and skip the brief and thus incomplete review of several well-known and otherwise comprehensively reviewed CPSs. This space might be used to discuss the gentic testing strategy and other critical issues mentioned above.

Specific

Page 2, line 57: In the context of CPS, SMN is not especially relevant in ped onc.

Table 1, Notes:

  • “only 5/15 genetic diagnosis (…) for the patient” has been mentioned for several studies, but it is not clear to me what it means.
  • Newman: I cannot understand “55% of germline (…) considered relevant”. 9.9% of what?

Page 7, line 197: please be more specific regarding the ovarian and testicular cancer (in PJS).

Page 7, line 211: I believe that Pheos are not a good example as their origin is quite heterogenous and the others missed to include SDHx genes here.

Page 7, second-to-last paragraph: Please also include FMTC when discussing MEN2A.

Page 9, first paragraph: In case of leukemia predisposition, I think it is not right to say that most genes are really associated with only a single subtype of hem cancer.

Page 9, line 304: I only know SBDS as a cause of SDS.

Page 9, second paragraph: it might have also been worth to mention the linked immunodeficiency seen in some patients with GATA2 deficiency.

I am not sure how relevant Table 2 is in general. However, I would revise “morphological examination”. I think “phenotypical” would be more appropriate. Regarding JMC: The questionnaire also includes kind of a tumor-specific algorithm.

Page 12, line 393ff, reference 83: The MIPOGG utility in predicting SMN is weak if one addresses the positive and negative predictive value of the app.

Looking at the test/tool performance, sensitivity and specificity is fine, but what means sensibility in this regard (main text and table 3).

Table 3: Demirsoy et al.: The authors state in Table 2 that FH is part of JC which is correct; thus, it doesn’t make sense to state “JC + cancer family history (…)” and conclude that FH can increase the frequency of genetic counselling referrals.

I would recommend to include the missing (known) causality in the sequencing all cons.

In the given pdf, there is no section heading 5 and 6?

Minor notes/comments

Abstract, line 30: it is not entirely clear to me what genetic evaluation and testing vs. genetic testing mean.

Introduction: Focusing on children, I think it would be more appropriate to use a childhood genturis example rather than HBOC. In line 52, page 2: please distinguish between past medical history and family history, since FH might be suspicious in the discussed setting of CPS w/o prototypical phenotypic features.

Page 2, line 60: I think it should be clear how to perform genetic testing. The question is what to test – also keeping in mind to address copy number and, if applicable, non-coding regions.

Please use proper HGNC protein terminology: RAS signaling pathway instead of Ras; TP53 protein instead of p53 protein.

The list of genetic causes re BWS is incomplete. I would recommend to include an “e.g.” – see page 6, line 156f. In line 158, chromosomal localization should be revised to “11p15.4p15.5” in accordance to ISCN. Same line, what is meant with “affected patients”.

What is a therapeutic program in VHL syndrome (page 6, line 181f).

Please be precise re 40% of RB – all, unilateral, bilateral and explain trilateral RB (first paragraph, page 7).

When reviewing some CPS, the authors specifically state the SMN risk for RB patients – but this is not specific for RB.

Page 15, line 422: Please add “heterozygous” to the “carriers”.

Page 15, line 431: Preimplantation diagnostics might be relevant in some countries too.

Please carefully revise English language.

Reviewer 2 Report

Suggestions:

  1. To replace in the text neoplasms with malignancies - more used in pediatric population
  2. To add in the keywords: recognition - that is the aim of a tools usage
  3. Line 42: "in the literature In the beginning" - small letter, should be in
  4. Line 45:  "relies mostly on history and physical examination" replace with - relies mostly on personal and family history and physical examination
  5.  Line 47: instead of "allowing for an accurate genetic", should be allowing an accurate genetic... 
  6. Line 50: ovarian cancer instead of ovary cancer
  7. Line 52 - same comment as for line 45
  8. Lines from 54-68 - space before references and further in the text, as well
  9. Line 162 - instead of adrenal carcinoma, adrenocortical carcinoma (ACC)
  10. Lines 182/182: instead of therapeutical program - specific treatment modalities.
  11. Line 202: instead of malifnancies - malignancies
  12. Line 204: instead of Also CNS tumors can be associated - CNS tumors can also be associated
  13. Line 231 - Hematopoietic Cell Transplantation: small letters
  14. Line 235 - instead of "might increase the genotoxicity of treatments" may increase the treatment related genotoxicity  
  15. Line 239: instead of head and neck cancer - head and neck cancers
  16. Line 239: Instead of rarer histotype - less frequent pathohistological sybtypes
  17. Line 249: instead of "to standard doses of chemotherapy" - standard doses of certain cytotoxic drugs. 
  18. Line 281/282: instead of "This probably happened because of the relatively easy accessibility of tumor samples for this kind of malignancies (i.e., a blood sample) - Easy accessibility of malignant cells facilitated the research and therapy design  and is currently...
  19. Line 292: instead of little - minor
  20. Line 301 instead of high grade - high risk patients
  21. Line 349 : "type and number of tumors" - type and number of tumors and its age of occurrence
  22. Line 405: instead of "were missed by both tools" - were missed by usage of both tools
  23. Line 428: instead of "showed a younger brother who is also affected. showed a younger brother who is also affected. The brother is currently undergoing periodic surveillance with regular clinical and hematological evaluations" - revealed the same genetic aberration in a younger brother, who is currently in undergoing the surveillance according to the most current recommendations. 
  24. Line 430: instead of "have other children" - have more children
  25. Line 464: instead of should  - would

I would add to the conclusion the significance of pre- and post- test counselling, when such an genetic investigation is done or is planned. 

Reviewer 3 Report

Dear Editor-in-chief,

the paper I had the pleasure to review is a review article by Rossini and colleagues reporting the diagnostic strategies and algorithms for investigating CPS in children with malignancies. The topic is interesting and appropriate for the journal. Contents are updated and correctly reported. I believe this is a good contribution to the field.

I would like to recommend some revisions before considering for publication:

  • Chapters are quite long and it is quite difficult to rapidly search for information in the text. I would suggest dividing the sections into shorter sub-sections.
  • Authors on Page 2 Line 60 reported that: "no consensus about when and how to perform germline genetic diagnostic studies in children with cancer" has been reported. While it is certainly true that no shared international guidelines are available, it is worth mentioning that national recommendations for specific cancers are available as in the Authors' state.
  • Considering monogenic mutation harboring increased risk for hematological malignancies I would suggest adding also novel genes such as ERCC6L2 (examples can be found in PMID: 30936069) and CBL (examples can be found in PMID: 35159106).
  • Considering the IEI harboring a higher risk of cancer I would suggest adding the already cited GATA2. Actually, the immunological phenotype of GATA2 mutations is peculiar and actually under investigation and should be mentioned (e.g. PMID: 25707267, 33066218, 24227816).
  • There are some spelling errors (e.g.; Pag 7 line 20 "malifnancies" instead of malignancies; Tabel 2 "tunor" instead of tumor). I would recommend the Authors carefully check all the text to correct these errors.
  • In the caption of Table 2, MIPOGG isn't defined. I would recommend adding the definition.